# PROB2VEC: MATHEMATICAL SEMANTIC EMBEDDING FOR PROBLEM RETRIEVAL IN ADAPTIVE TUTORING

## ABSTRACT

We propose a new application of embedding techniques to problem retrieval in adaptive tutoring. The objective is to retrieve problems similar in mathematical concepts. There are two challenges: First, like sentences, problems helpful to tutoring are never exactly the same in terms of the underlying concepts. Instead, good problems mix concepts in innovative ways, while still displaying continuity in their relationships. Second, it is difficult for humans to determine a similarity score consistent across a large enough training set. We propose a hierarchical problem embedding algorithm, called Prob2Vec, that consists of abstraction and embedding steps. Prob2Vec achieves 96.88% accuracy on a problem similarity test, in contrast to 75% from directly applying state-of-the-art sentence embedding methods. It is surprising that Prob2Vec is able to distinguish very fine-grained differences among problems, an ability humans need time and effort to acquire. In addition, the sub-problem of concept labeling with imbalanced training data set is interesting in its own right. It is a multi-label problem suffering from dimensionality explosion, which we propose ways to ameliorate. We propose the novel negative pre-training algorithm that dramatically reduces false negative and positive ratios for classification, using an imbalanced training data set.

## 1 INTRODUCTION

The traditional teaching methods that are widely used at universities for science, technology, engineering, and mathematic (STEM) courses do not take different abilities of learners into account. Instead, they provide learners with a fixed set of textbooks and homework problems. This ignorance of learners' prior background knowledge, pace of learning, various preferences, and learning goals in current education system can cause tremendous pain and discouragement for those who do not keep pace with this inefficient system Chen (2008); Brusilovsky & Vassileva (2003); Chen et al. (2005); Hübscher (2000); Weber & Specht (1997). Hence, e-learning methods are given considerable attention in an effort to personalize the learning process by providing learners with optimal and adaptive curriculum sequences. Over the years, many web-based tools have emerged to adaptively recommend problems to learners based on courseware difficulty. These tools tune the difficulty level of the recommended problems for learners and push them to learn by gradually increasing the difficulty level of recommended problems on a specific concept. The downside of such methods is that they do not take the concept continuity and mixture of concepts into account, but focus on the difficulty level of single concepts. Note that a learner who knows every individual concept does not necessarily have the ability to bring all of them together for solving a realistic problem on a mixture of concepts. As a result, the recommender system needs to know similarity/dissimilarity of problems with mixture of concepts to respond to learners' performance more effectively as described in the next paragraph, which is something that is missing in the literature and needs more attention.

Since it is difficult for humans to determine a similarity score consistent across a large enough training set, it is not feasible to simply apply supervised methods to learn a similarity score for problems. In order to take difficulty, continuity, and mixture of concepts into account for similarity score used in a personalized problem recommender system in an adaptive practice, we propose to use a proper numerical representation of problems on mixture of concepts equipped with a similarity measure. By virtue of vector representations for a set of problems on both single and mixture of concepts (problem embedding) that capture similarity of problems, learners' performance on a problem can be projected onto other problems. As we see in this paper, creating a proper problem representation

that captures mathematical similarity of problems is a challenging task, where baseline text representation methods and their refined versions fail to work. Although the state-of-the-art methods for phrase/sentence/paragraph representation are doing a great job for general purposes, their shortcoming in our application is that they take lexical and semantic similarity of words into account, which is totally invalid when dealing with text related to math or any other special topic. The words or even subject-related keywords of problems are not completely informative and cannot contribute to embedding of math problems on their own; as a result, the similarity of two problems is not highly correlated with the wording of the problems. Hence, baseline methods perform poorly on the problem similarity detection test in problem recommender application.

We find that instead of words or even subject-related keywords, conceptual ideas behind the problems determine their identity. The conceptual particles (concepts) of problems are mostly not directly mentioned in problem wording, but there can be footprints of them in problems. Since problem wording does not capture the similarity of problems, we propose an alternative hierarchical approach called Prob2Vec consisting of an abstraction and an embedding step. The abstraction step projects a problem to a set of concepts. The idea is that there exists a concept space with a reasonable dimension $N$, with $N$ ranging from tens to a hundred, that can represent a much larger variety of problems of order $O(2^N)$. Each variety can be sparsely inhabited, with some concept combination having only one problem. This is because making problems is itself a creative process: The more innovative a problem is, the less likely it has exactly the same concept combination as another problem. The explicit representation of problems using concepts also enables state-dependent similarity computation, which we will explore in future work. The embedding step constructs a vector representation of the problems based on concept cooccurrence. Like sentence embedding, not only does it capture the common concepts between problems, but also the continuity among concepts. The proposed Prob2Vec algorithm achieves 96.88% accuracy on a problem similarity test, where human experts are asked to label the relative similarity among each triplet of problems. In contrast, the best of the existing methods, which directly applies sentence embedding, achieves 75% accuracy. It is surprising that Prob2Vec is able to distinguish very fine-grained differences among problems, as the problems in some triplets are highly similar to each other, and only humans with extensive training in the subject are able to identify their relative order of similarity. The problem embedding obtained from Prob2Vec is being used in the recommender system of an e-learning tool for an undergraduate probability course for four semesters with successful results on hundreds of students, specially benefiting minorities who tend to be more isolated in the current education system.

In addition, the sub-problem of concept labeling in the abstraction step is interesting in its own right. It is a multi-label problem suffering from dimensionality explosion, as there can be as many as $2^N$ problem types. This results in two challenges: First, there are very few problems for some types, hence a direct classification on $2^N$ classes suffers from a severe lack of data. Second, per-concept classification suffers from imbalance of training samples and needs a very small per-concept false positive in order to achieve a reasonable per-problem false positive. We propose pre-training of the neural network with negative samples (negative pre-training) that beats a similar idea to one-shot learning Fei-Fei et al. (2006), where the neural network is pre-trained on classification of other concepts to have a warm start on classification of the concept of interest (transfer learning).

## 1.1 RELATED WORK

**Embedding applications:** the success of simple and low-cost word embedding technique using the well-known neural network (NN) based word embedding method, Word2Vec by Mikolov et al. (2013b;a), compared to expensive natural language processing methods, has motivated researchers to use embedding methods for many other areas. As examples, Doc2Vec Lau & Baldwin (2016), Paper2Vec Tian & Zhuo (2017); Ganguly & Pudi (2017), Gene2Vec Cox, Graph2Vec Narayanan et al. (2017), Like2Vec, Follower2Vec and many more share the same techniques originally proposed for word embedding with modifications based on their domains, e.g. see Ganguly & Pudi (2017); Grover & Leskovec (2016); Barkan & Koenigstein (2016); Narayanan et al. (2016); Dhingra et al. (2016); Ma et al. (2016); Ring et al. (2017); Dong et al. (2017); Ribeiro et al. (2017); Chen et al. (2017); Chung et al. (2016); Niu et al. (2015); Vasile et al. (2016); Lin et al. (2017); Ristoski & Paulheim (2016); Melamud et al. (2016); Shi et al. (2018); Liu et al. (2018). In this work, we propose a new application for problem embedding for personalized problem recommendation.

**Word/Phrase/Sentence/Paragraph embedding:** the prior works on word embedding include learning a distributed representation for words by Bengio et al. (2003), multi-task training of a

convolutional neural network using weight-sharing by Collobert & Weston (2008), the continuous Skip-gram model by Mikolov et al. (2013b), and the low rank representation of word-word co-occurrence matrix by Deerwester et al. (1990); Pennington et al. (2014). Previous works on phrase/sentence/paragraph embedding include vector composition that is operationalized in terms of additive and multiplicative functions by Mitchell & Lapata (2008; 2010); Blacoe & Lapata (2012), uniform averaging for short phrases by Mikolov et al. (2013b), supervised and unsupervised recursive autoencoder defined on syntactic trees by Socher et al. (2011; 2014), training an encoder-decoder model to reconstruct surrounding sentences of an encoded passage by Kiros et al. (2015), modeling sentences by long short-term memory (LSTM) neural network by Tai et al. (2015) and convolutional neural networks by Blunsom et al. (2014), and the weighted average of the words in addition to a modification using PCA/SVD by Arora et al. (2016). The simple-tough-to-beat method proposed by Arora et al. (2016) beats all the previous methods and is the baseline in text embedding.

The rest of the paper is outlined as follows. The description of the data set for which we do problem embedding in addition to our proposed Prob2Vec method for problem embedding are presented in section 2. Negative pre-training for NN-based concept extraction is proposed in section 3. Section 4 describes the setup for similarity detection test for problem embedding, evaluates the performance of our proposed Prob2Vec method versus baselines, and presents the results on the negative pre-training method. Section 5 concludes the paper with a discussion on opportunities for future work.

## 2 PROB2VEC: CONCEPT-BASED PROBLEM EMBEDDING

Consider a set of $M$ problems $\mathcal{P} = \{P_1, P_2, \cdots, P_M\}$ for an undergraduate probability course, where each problem can be on a single or mixture of concepts among the set of all $N$ concepts $\mathcal{C} = \{C_1, C_2, \cdots, C_N\}$. Note that these concepts are different from keywords of problem wordings and are not originally included in problems, but labeling problems with concepts is a contribution of this work that is proposed for achieving a proper problem representation. Instead, problems are made of words from the set $\mathcal{W} = \{W_1, W_2, \cdots, W_L\}$; i.e. $P_i = \mathcal{W}_i$ for $1 \leq i \leq M$, where $\mathcal{W}_i \subset \mathcal{W}^{|\mathcal{W}_i|}$. In the following subsection, we propose the Prob2Vec method for problem embedding that uses an automated rule-based concept extractor, which relieves reliance on human labeling and annotation for problem concepts.

### 2.1 PROB2VEC PROBLEM EMBEDDING

As shown in section 4, using the set of words $\mathcal{W}_i|_{i=1}^M$ or even a subset of keywords to represent problems, text embedding baselines fail to achieve high accuracy in similarity detection task for triplets of problems. In the keyword-based approach, all redundant words of problems are ignored and the subject-related and informative words such as binomial, random variable, etc., are kept. However, since the conceptual ideas behind problems are not necessarily mapped into normal and mathematical words used in problems, even the keyword-based approach fails to work well in the similarity detection task that is explained in section 4. Alternatively, we propose a hierarchical method consisting of abstraction and embedding steps that generates a precise embedding for problems that is completely automated. The block diagram of the proposed Prob2Vec method is depicted in figure 1.

(i) **Abstraction step:** similarity among mathematical problems is not captured by the wording of problems; instead, it is determined by the abstraction of the problems. Learners who have difficulty solving mathematical problems mostly lack the ability to do abstraction and relate problems with appropriate concepts. Instead, they try to remember procedure-based rules to fit problems in them and use their memory to solve them, which does not apply to solving hard problems on mixture of concepts. We observe the same pattern in problem representation; i.e. problem statements do not necessarily determine their identity, instead abstraction of problems by mapping them into representative concepts moves problem embedding from lexical similarity to conceptual similarity. The concepts of a problem are not mostly mentioned directly in its text, but there can be footmarks of concepts in problems. A professor and two experienced teaching assistants are asked to formulate rule-based mappings from footmarks to concepts for automation of concept extraction. As an example, the rule for labeling a problem with concept "nchoosek" is `\\\\choose|\\\\binom|\\\\frac\{\s*\w+\!\s*\}\{\s*\w+\!\s*\\\\times\s*\w+\!\s*\}`. By applying the rule-based concept extractor to problems, we have a new representation for problems in concept space instead of word space; i.e. $P_i \equiv \mathcal{C}_i$ for $1 \leq i \leq M$, where $\mathcal{C}_i \subset \mathcal{C}$.

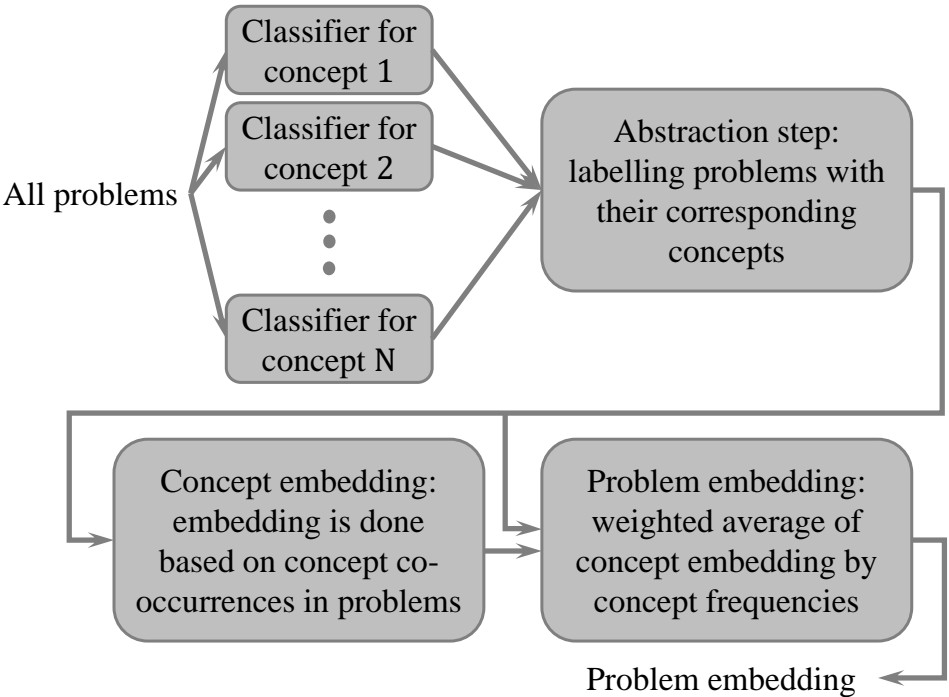

Figure 1: Block diagram of the hierarchical Prob2Vec method with abstraction and embedding steps.

(ii) **Embedding step:** a method similar to Skip-gram in Word2Vec is used for concept embedding. The high-level insight of Skip-gram is that a neural network with a single hidden layer is trained, where its output relates to how likely it is to have each concept co-occurred in a problem with the input concept. As an example, if concept "law-of-total-probability" of a problem is input of the neural network, we expect the neural network to state it more likely to have concept "conditional-probability" in the problem than for unrelated concepts like "Poisson-process". However, the neural network is not used for this task, but the goal is to use weights of the hidden layer for embedding. Recall the set of all concepts as $\{C_1, C_2, \cdots, C_N\}$, where a problem is typically labeled with a few of them. Consider one-hot coding from concepts to real-valued vectors of size $N$ that are used for training of the neural network, where the element corresponding to a concept is one and all other elements are zero. We consider 10 neurons in hidden layer with no activation functions (so the embedded concept vectors have 10 features) and $N$ neurons in the output that form a softmax regression classifier. In order to clarify on input-output pair of neural network, assume a problem that has a set of concepts $\{C_1, C_2, C_5\}$. The neural network is trained on all pairs $(C_1, C_2), (C_1, C_5), (C_2, C_1), (C_2, C_5), (C_5, C_1)$, and $(C_5, C_2)$, where the one-hot code of the first element of a pair is the input and the one-hot code of the second element of a pair is the output of the neural network in the training phase. Hence, the neural network is trained over $\sum_{i=1}^{M} |\mathcal{C}_i| \times (|\mathcal{C}_i| - 1)$ number of training data. This way, the neural network learns the statistic from the number of times that a pair is fed into it (the neural network is probably fed with more training pairs of ("law-of-total-probability", "conditional-probability") than the pair ("law-of-total-probability", "Poisson-process")). Note that during training phase, input and output are one-hot vectors representing the input and output concepts, but after training when using the neural network, given a one-hot input vector, the output is a probability distribution on the set of all concepts. Finally, since input concepts are coded as one-hot codes, rows of hidden layer weight matrix, which is of size $N$ by 10, are concept vectors (concept embedding) which we are really after. Denoting embedding of concept $c \in \mathcal{C}$ by $E(c)$, problem embedding denoted by $E_i$ for problem $P_i$ is obtained as follows:

$$E_i = \frac{1}{|\mathcal{C}_i|} \sum_{c \in \mathcal{C}_i} \frac{1}{f_c} \cdot E(c), \tag{1}$$

where $f_c$ is frequency of concept $c \in \mathcal{C}$ in our data set of $M = 635$ problems. Concept embedding is scaled by its corresponding frequency to penalize concepts with high frequency that are not as informative as low frequency ones. For example, concept "pmf" is less informative than concept "ML-parameter-estimation". Given problem embedding, similarity between two problems is defined as cosine of the angle between corresponding vector representations.

Remark.    We choose rule-based concept extractor for the abstraction step over any supervised/unsupervised classification methods for concept extraction because of two main reasons. First, there is a limited number of problems for most concepts as few as a couple of problems which makes any supervised classification method inapplicable due to lack of training data. Second, there are $N$ concepts, so potentially there can be $2^N - 1$ categories of problems which makes classification challenging for any supervised or unsupervised methods. Consider the maximum number of concepts in a problem to be $k$, then there are on the order of $O(N^k)$ categories of problems. Even if we consider possessing acceptable number of problems for each of the $O(N^k)$ categories, false positive needs to be on the order of $O(\frac{1}{N^k})$ so that the number of false positives for each category be in the order of $O(1)$. Given that there are $N = 96$ concepts, utilizing a supervised or unsupervised approach to achieve such a low false positive is not feasible. Exploiting the rule-based classifier for problem labeling though, we succeed to achieve as low as average $0.98\%$ false positive and average $9.17\%$ false negative for all concepts, where problem concepts annotated by experts are considered to be ground truth. Although $100.00\%$ accuracy is observed in similarity detection test when utilizing problem concepts annotated by experts, those concepts are not necessarily the global optimum ground truth or the only one. Thinking of increasing accuracy in similarity detection task as an optimization problem, there is not necessarily a unique global optimum for problem concepts that lead to good performance. Hence, not having a very low false negative for rule-based does not necessarily mean that such labels are not close to a local/global optimum problem concepts. In fact, rule-based extracted concepts achieve a high $96.88\%$ accuracy on a similarity test as is mentioned in section 4.

## 3  NEGATIVE PRE-TRAINING FOR IMBALANCED DATA SETS

For the purpose of problem embedding, Prob2Vec discussed in section 2.1 with a rule-based concept extractor has an acceptable performance. Here, a NN-based concept extractor is proposed that can be a complement to the rule-based version, but we mainly study it to propose our novel negative pre-training method for reducing false negative and positive ratios for concept extraction with an imbalanced training data set. Negative pre-training outperforms a similar method to one-shot learning (transfer learning) as data level classification algorithms to tackle imbalanced training data sets.

The setup for concept extractor using neural networks without any snooping of human knowledge is presented first, then we propose some tricks for reducing false negative and positive ratios. The neural network used for concept extraction has an embedding layer with linear perceptrons followed by two layers of perceptrons with sigmoid activation function, and an output layer with a single perceptron with sigmoid classifier. For common words in Glove and our data set, embedding layer is initialized by Glove Pennington et al. (2014), but for words in our data set that are not in Glove, the weights are initialized according to a uniform distribution over $[-1, 1]$. The embedding size is considered to be 300 and each of the other two layers have 60 perceptrons, followed by output which has a single perceptron, indicating if a concept is in the input problem or not. Note that for each concept, a separate neural network is trained. The issue with training a single neural network for all concepts is the imbalanced number of positive and negative samples for each concept. A concept is only present in a few of $M = 635$ problems, so having a single neural network, too many negative samples for each concept are fed into it, dramatically increasing false negatives.

In the following, some tricks used for training of the above naive NN-based concept extractor are presented that reduce FN and FP ratios by at least $43.67\%$ and up to $76.51\%$ compared to using down sampling, which is a standard approach for training on imbalanced training data sets. The main challenge in obtaining low FN and FP is that as few as 12 problems are labeled by a concept, which makes training of neural network challenging by having 12 positive and 623 negative samples.

(a) **Negative pre-training**: few of $M = 635$ problems are labeled with a specific concept, e.g. 12 problems have concept "hypothesis-MAP". Hence, few positive samples and many negative

samples are provided in our data set for training of a NN-based concept extractor for a specific concept. A neural network cannot obviously be trained on an imbalanced set where all negative samples are mixed with few positive ones or FN increases dramatically. Instead, we propose two phases of training for concept $C_i$. Consider $\mathcal{P}_i = \{P_j : C_i \in \mathcal{C}_j\}$ and $\mathcal{N}_i = \{P_j : C_i \notin \mathcal{C}_j\}$, where $|\mathcal{N}_i| \gg |\mathcal{P}_i|$. Let $\overline{\mathcal{N}}_i \subset \mathcal{N}_i$, where $|\overline{\mathcal{N}}_i| = |\mathcal{P}_i|$. In the first phase, neural network is pre-trained on a pure set of negative samples, $\mathcal{N}_i \setminus \overline{\mathcal{N}}_i$, where the trained neural network is used as a warm start for the second phase. In the second phase of training, neural network is trained on a balanced mixture of positive and negative samples, $\mathcal{P}_i \cup \overline{\mathcal{N}}_i$. Utilizing negative pre-training, we take advantage of negative samples in training, and not only does FN not increase, but we get an overall lower FN and FP compared to down sampling. Due to curse of dimensionality, neural network learns a good portion of the structure of negative samples in the first phase of negative pre-training that provides us with a warm start for the second phase.

(b) One-shot learning (transfer learning), Fei-Fei et al. (2006): in the first phase of training, the neural network is first trained on classification of bags of problems with equal number of negative and positive samples of concepts that are not of interest, $\mathcal{P}_j \cup \overline{\mathcal{N}}_j\big|_{j \neq i}$. Then, the trained neural network is used as a warm start in the second training phase for classification of the concept of interest on a balanced set, $\mathcal{P}_i \cup \overline{\mathcal{N}}_i$.

(c) Word selection: due to limited number of positive training samples, a neural network cannot tune any number of parameters to find important features of problems. Moreover, as a rule of thumb, the fewer parameters the neural network has, the less it is vulnerable to over-fitting and the faster it can converge to an acceptable classifier. To this end, an expert TA is asked to select informative words out of total 2242 words that are originally used for input of neural network, where this process took less than an hour. The redundant words in problems are omitted and only those among 208 selected words related to probability are kept in each problem, which reduces size of embedding matrix from $2242 \times 300$ to $208 \times 300$ and inputs more informative features to neural network. In section 4, it is shown that FP and FN ratios are reduced under this trick by at least $25.33\%$ and up to $61.34\%$, which is an indication that selected words are more representative than the original ones. These selected words have been used in problem embedding for modified versions of baselines in section 4 as evidence that even keyword-based versions of embedding baselines do not capture similarity of problems.

## 4 EXPERIMENTS AND RESULTS

For evaluation of different problem embedding methods, a ground truth on similarity of problems is needed. To this end, four TAs are asked to select random triplets of problems, say $(A, B, C) \in \mathcal{P}^3$ with $A \neq B \neq C$, and order them so that problem $A$ is more similar to $B$ than $C$; i.e. if the similarity between two problems is denoted by $sim(.,.)$, we have $sim(A, B) > sim(A, C)$. Finally, a head TA brings results into a consensus and chooses 64 triplets of problems. Note that the set of all $M = 635$ problems are divided into 26 modules, where each module is on a specific topic, e.g. hypothesis testing, central limit theorem, and so on. The three problems of a triplet are determined to be in the same module, so they are already on the same topic that makes similarity detection task challenging. As evidence, the similarity gap histogram for the 64 triplets of problems, $sim(A, B) - sim(A, C)$, according to expert annotation for problem concepts and Skip-gram-based problem embedding, is shown in figure 2. It should be noted that the mentioned problem embedding is empirically proven to have the highest accuracy of $100.00\%$ in our similarity detection test. The expert annotation for problem concepts are done by an experienced TA unaware of problem embedding project, so no bias is brought to concept labeling process. The similarity gap histogram depicts it well that similarity detection for the 64 triplets of problems is challenging due to skewedness of triplets in first bins.

Prob2Vec is compared with different baseline text embedding methods in terms of the accuracy in determining the more similar problem of the second and third to the first one of a triplet. The experimental results are reported in table 1. The baselines are mainly categorized into three groups as 1- Glove-based problem embedding that is derived by taking the uniform (or weighted with word frequencies) average of Glove word embedding, where the average can be taken over all words of a problem or some representative words of the problem. 2- Arora et al. (2016) suggest to remove the first singular vector from Glove-based problem embedding, where that singular vector corresponds to syntactic information and common words. 3- SVD-based problem embedding that has the same

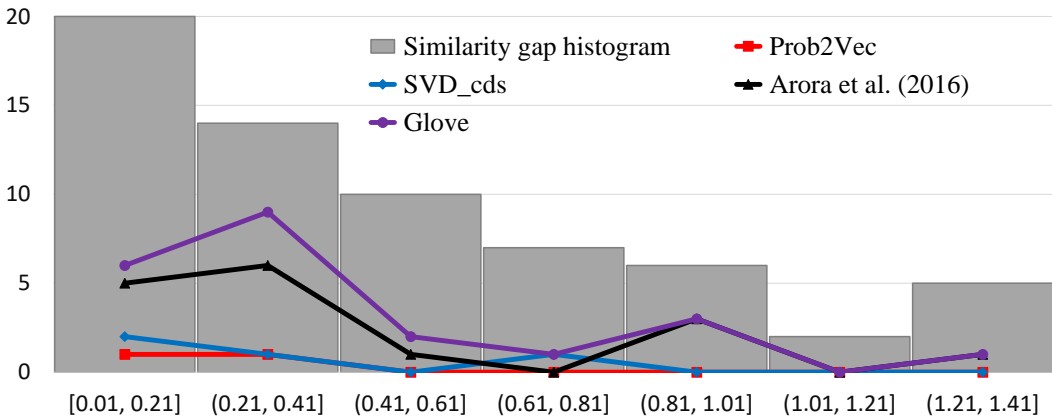

Figure 2: The similarity gap histogram of the 64 triplets of problems, and the number of errors per bin on similarity detection test for different embedding methods.

| concepts
method | annotated concepts | rule-based concepts |
|---|---|---|
| **Prob2Vec** | **100.00**% | **96.88**% |
| SVD_sub | 93.75% | 87.50% |
| SVD_shifted | 85.94% | 84.38% |
| SVD_cds | 93.75% | 93.75% |
| SVD_wandc | 93.75% | 89.06% |
| SVD_eig | 92.19% | 87.50% |
| | all words | selected words |
| Arora et al. (2016) | 75.00% | 75.00% |
| | uniform average | weighted average |
| Glove (all words) | 64.06% | 56.25% |
| Glove (selected words) | 65.62% | 39.06% |

Table 1: Accuracy of different embedding methods for similarity detection test.

hierarchical approach as Prob2Vec, but concept embedding in the second step is done based on SVD decomposition of the concept co-occurrence matrix Levy et al. (2015). The details on baseline methods can be found in appendix A. The number of errors that the method with the best performance in each of the above categories makes in different bins of the similarity gap are shown in figure 2. For example, there are 20 triplets with similarity gap in the range $[0.01, 0.21]$ and the best Glove-based method makes six errors out of these 20 triplets in the similarity detection test. According to table 1, the best Glove-based method is taking uniform average of embedding of selected words.

Interesting patterns on concept continuity and similarity are observed from Prob2Vec concept embedding where two of them are shown in table 2. As other examples, it is observed that the most similar concept to function-RV is CDF, where function-RV refers to finding the distribution of a function of a random variable. As a TA of probability course for three years, most students do not have any clues where to start on problems for function-RV, and we always tell them to start with finding CDF of the function of random variable. It is worthy to see that NN-based concept embedding can capture such relation between concepts in seconds with small number of training samples while a human that is trained over the whole semester at university is mostly clueless where to start. We further observe the ML-parameter-E concept to be most related to the concept differentiation, where ML-parameter-E refers to maximum likelihood (ML) parameter estimation. Again, students do not get this relation for a while and they need a lot of training to get the association of ML-parameter-E with differentiation of likelihood of observation to find ML parameter estimation. As another example, Bayes-formula is most similar to law-of-total-probability and the list goes on.

| Similar concepts / Concept | 1 | 2 | 3 | 4 |
|---|---|---|---|---|
| prob-error | prob-miss 0.9766 | prob-false 0.9699 | hypothesis-MAP 0.9661 | hypothesis-ML 0.9122 |
| mutually-exclusive | partition 0.9078 | event 0.9056 | set-complement 0.8610 | set-intersection 0.8270 |

Table 2: Concept continuity for concepts "prob-error" and "mutually-exclusive".

| | concepts | 1 | 2 | 3 | 4 | 5 |
|---|---|---|---|---|---|---|
| down sampling | FN | 9.09% | 9.16% | 7.80% | 11.11% | 10.45% |
| | FP | 8.17% | 10.45% | 8.49% | 11.95% | 10.70% |
| down sampling + word selection | FN | 5.74% | 6.84% | 4.67% | 5.19% | 5.47% |
| | FP | 4.76% | 5.95% | 3.71% | 4.62% | 5.02% |
| negative pre-training | FN | 5.56% | 8.15% | 3.55% | 6.71% | 5.68% |
| | FP | 6.20% | 8.88% | 5.08% | 5.32% | 6.79% |
| one-shot learning | FN | 9.92% | 9.79% | 5.22% | 8.96% | 9.69% |
| | FP | 8.55% | 6.18% | 7.89% | 6.77% | 6.44% |
| **word selection + neg. pre-training** | **FN** | **3.38%** | **5.16%** | **3.01%** | **2.61%** | **5.32%** |
| | **FP** | **3.80%** | **4.92%** | **3.18%** | **4.90%** | **4.13%** |
| word selection + one-shot learning | FN | 4.84% | 6.46% | 4.62% | 4.91% | 5.61% |
| | FP | 4.35% | 6.78% | 3.51% | 5.04% | 5.69% |
| one-shot learning + neg. pre-training | FN | 6.67% | 8.42% | 5.26% | 6.82% | 6.98% |
| | FP | 8.93% | 8.76% | 6.21% | 6.49% | 6.39% |
| word sel. + one-shot + neg. pre. | FN | 3.82% | 6.93% | 3.57% | 2.41% | 3.33% |
| | FP | 4.90% | 5.03% | 3.57% | 4.49% | 4.36% |

Table 3: False negative and positive ratios of NN-based concept extraction for 1: "sample-space", 2: "Covariance", 3: "joint-Gaussian", 4: "hypothesis-MAP", 5: "hypothesis-ML".

Comparison of negative pre-training, one-shot learning, word selection, down sampling, and combination of these methods applied to training process of the NN-based concept extractor is presented in table 3 for five concepts. In order to find the empirical false negative and positive ratios for each combination of methods in table 3, training and cross validation are done for 100 rounds on different random training and test samples, and false negative and positive ratios are averaged in the 100 rounds. As an example, the set $\overline{\mathcal{N}}_i \subset \mathcal{N}_i$ and test set are randomly selected in each of the 100 rounds for negative pre-training method, then the false negative and positive ratios of the trained neural network on the 100 instances are averaged. Employing the combination of word selection and negative pre-training reduces false negative and positive ratios by at least 43.67% and up to 76.51% compared to the naive down sampling method. For some concepts, the combination of word selection, one-shot learning, and negative pre-training results in a slightly lower false negative and positive ratios than the combination of word selection and negative pre-training. However, investigating the whole table, one finds out that word selection and negative pre-training are the causes for reducing false negative and positive ratios. It is of interest that NN-based approach can reduce FN for the concept "event" to 5.11% with FP of 6.06%, where rule-based has FN of 35.71% with FP of 5.31%.

## 5 CONCLUSION AND FUTURE WORK

A hierarchical embedding method called Prob2Vec for subject specific text is proposed in this paper. Prob2Vec is empirically proved to outperform baselines by more than 20% in a properly validated similarity detection test on triplets of problems. The Prob2Vec embedding vectors for problems are being used in the recommender system of an e-learning tool for an undergraduate probability course for four semesters. We also propose negative pre-training for training with imbalanced data sets to decrease false negatives and positives. As future work, we plan on using graphical models along with problem embedding vectors to more precisely evaluate the strengths and weaknesses of students on single and mixture of concepts to do problem recommendation in a more effective way.

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

## A    DETAILS OF EXPERIMENTAL SETTING

The proposed Prob2Vec method is compared with the following text embedding methods.

(i) 1- **Glove:** referring to literature for text embedding, the following are probably the most primitive approaches for problem embedding denoted by $E_i$ for problem $P_i$:

$$E_i = \frac{1}{|\mathcal{W}_i|} \sum_{w \in \mathcal{W}_i} \widehat{E}_w, \tag{2}$$

$$E_i = \frac{1}{|\mathcal{W}_i|} \sum_{w \in \mathcal{W}_i} \frac{1}{\widehat{f}_w} \cdot \widehat{E}_w, \tag{3}$$

where $\widehat{E}_w$ and $\widehat{f}_w$ are Glove embedding and frequency of word $w \in \mathcal{W}$ in our data set of $M = 635$ problems, respectively. We can either consider all words of a problem, $\mathcal{W}_i$, or use selected subject-related keywords discussed in section 3 in the above embedding formulas. The selected keywords are logical choices since we get lower false positives and negatives in concept extraction by using these keywords instead of all words as shown in section 4.

(ii) 2- **Arora et al. (2016):** as an improvement on the previous raw method, Arora et al. (2016) find out that top singular vectors of text embedding seem to roughly correspond to syntactic information and common words. Hence, they propose to exclude the first singular vector from sentence (problem) embedding in the following way. Given word embedding computed using

one of popular methods, $\left\{ \widehat{E}_w : w \in \mathcal{W} \right\}$, where we use Glove, problem embedding for $P_i$ that is denoted by $E_i$ is computed as follows:

$$\overline{E}_i = \frac{1}{|\mathcal{W}_i|} \sum_{w \in \mathcal{W}_i} \frac{a}{a + p(w)} \widehat{E}_w, \quad \text{for } 1 \le i \le M,$$

$$E_i = \overline{E}_i - uu^\top \overline{E}_i, \quad \text{for } 1 \le i \le M,$$

where $u$ is the first principle component of $\left\{ \overline{E}_i : 1 \le i \le M \right\}$ and $a$ is a hyper-parameter which is claimed to result in best performance when $a = 10^{-3}$ to $a = 10^{-4}$. We tried different values for $a$ inside this interval and out of it and found $a = 10^{-5}$ and $a = 10^{-3}$ to best work for our data set when using all words and $a = 2 \times 10^{-2}$ to best work for when using selected words.

(iii) 3- **SVD:** using the same hierarchical approach as Prob2Vec, concept embedding in the second step can be done with an SVD-based method instead of the Skip-gram method as follows. Recall that the concept dictionary is denoted by $\{C_1, C_2, \cdots, C_N\}$, where each problem is labeled with a subset of these concepts. Let $N_c(C_i, C_j)$ for $i \ne j$ denote number of co-occurrences of concepts $C_i$ and $C_j$ in problems of data set; i.e. there are $N_c(C_i, C_j)$ number of problems that are labeled with both $C_i$ and $C_j$. The co-occurrence matrix is formed as follows:

$$M_c = \begin{bmatrix} 0 & N_c(C_1, C_2) & N_c(C_1, C_3) & \dots & N_c(C_1, C_N) \\ N_c(C_2, C_1) & 0 & N_c(C_2, C_3) & \dots & N_c(C_2, C_N) \\ \vdots & \vdots & \vdots & \ddots & \vdots \\ N_c(C_N, C_1) & N_c(C_N, C_2) & N_c(C_N, C_3) & \dots & 0 \end{bmatrix}.$$

The co-occurrence matrix is obviously symmetric with diagonal elements being zero and $M_c(i,j) = M_c(j,i) = N_c(C_i, C_j)$. By defining $D = \sum_{i=1}^N \sum_{j=1}^N M_c(i,j)$ and $w(i) = \frac{\sum_{j=1}^N M_c(i,j)}{D}$ for $1 \le i \le N$, the $PPMI$ matrix is constructed as follows:

$$PPMI(i,j) = \begin{cases} \log\left( \frac{M_c(i,j)}{D \cdot w(i) \cdot w(j)} \right), & \text{if } \frac{M_c(i,j)}{D \cdot w(i) \cdot w(j)} > 1 \text{ and } w(i) \cdot w(j) \ne 0, \\ 0, & \text{otherwise.} \end{cases}$$

The SVD decomposition of the $PPMI$ matrix is as $PPMI = USV$, where $U, S, V \in \mathbb{R}^{N \times N}$, and $S$ is a diagonal matrix. Denote embedding size of concepts by $d \le N$, and let $U_d$ be the first $d$ columns of matrix $U$, $S_d$ be a diagonal matrix with the first $d$ diagonal elements of diagonal matrix $S$, and $V_d$ be the first $d$ rows of matrix $V$. The followings are different variants of SVD-based concept embedding Levy et al. (2015):

- **eig**: embedding of $N$ concepts are given by $N$ rows of matrix $U_d$ that are of embedding length $d$.
- **wandc**: let $W = U_d S_d$ and $C = V_d^T$. Embedding of $N$ concepts are given by $N$ rows of $W + C$.
- **sub**: $N$ rows of $U_d S_d$ are embedding of $N$ concepts.
- **shifted**: the $PPMI$ matrix is defined in a slightly different way in this variant as follows:

$$PPMI(i,j) = \begin{cases} \log\left( \frac{M_c(i,j)}{D \cdot w(i) \cdot w(j)} \right), & \text{if } \frac{M_c(i,j)}{D \cdot w(i) \cdot w(j)} > k \text{ and } w(i) \cdot w(j) \ne 0, \\ 0, & \text{otherwise.} \end{cases}$$

  We choose $k = 5$ in our evaluations and calculate $U_d$ and $S_d$ based on the above $PPMI$ matrix as before. The $N$ rows of $U_d S_d$ are embedding of $N$ concepts.

- **cds**: the $PPMI$ matrix is defined in a slightly different way in this variant. Let $D_c = \sum_{i=1}^N \left( \sum_{j=1}^N M_c(i,j) \right)^{0.75}$ and $\overline{w}(i) = \frac{\left( \sum_{j=1}^N M_c(i,j) \right)^{0.75}}{D_c}$. The $PPMI$ matrix is then defined as follows:

$$PPMI(i,j) = \begin{cases} \log\left( \frac{M_c(i,j)}{D \cdot w(i) \cdot \overline{w}(j)} \right), & \text{if } \frac{M_c(i,j)}{D \cdot w(i) \cdot \overline{w}(j)} > 1 \text{ and } w(i) \cdot \overline{w}(j) \ne 0, \\ 0, & \text{otherwise.} \end{cases}$$

  Note that the $PPMI$ matrix is not necessarily symmetric in this case. By deriving $U_d$ and $S_d$ matrices as before, embedding of $N$ concepts are given by $N$ rows of $U_d S_d$.

