# OpenReview forum: "Prob2Vec: Mathematical Semantic Embedding for Problem Retrieval in Adaptive Tutoring"
_ICLR.cc/2019/Conference_

### Official Review · AnonReviewer1 · 2018-10-22
**Review of "Prob2Vec: Mathematical Semantic Embedding for Problem Retrieval in Adaptive Tutoring"**

**Rating:** 4
**Confidence:** 4

**Review:**

This paper proposes a new application of embedding techniques for mathematical problem retrieval in adaptive tutoring. The proposed method performs much better than baseline sentence embedding methods. Another contribution is on using negative pre-training to deal with an imbalanced training dataset.

To me this paper is just not good enough - the method essentially i) use "a professor and two teaching assistants" to build a "rule-based concept extractor" for problems, then ii) map problems into this "concept space" and simply treat them as words. There are several problems with this approach.

First, doing so does not touch the core of the proposed application. For tutoring applications, the most important thing is to select a problem that can help students improve; even if you can indeed select a problem that is the most similar to another problem, is it the best one to show a student? There are no evaluations on real students in the paper. Moreover, the main difference between math problems and other problems is that there are math expressions; I do not think that using words/concept labels only is enough without touching on the math expressions.

Second, the proposed method does not sound scalable - the use of a professor and two teaching assistants to construct the concept extractor, and the use of an expert TA to select a small set of informative words. I am not sure how this will generalize to a larger number of problem spanning many different domains.

I also had a hard time going through the paper - there aren't many details. Section 2.1 is where the method is proposed, yet most of the descriptions there are unclear. Without these details it is impossible to judge the novelty of the "rule-based concept extractor", which is the key technical innovation.

---

> ### Author Response · Authors · 2018-11-07
> **Response to Reviewer3**
>
>
> 1- We briefly mentioned the way problem embedding with similarity metric is used in the recommendation system in this work, but here is more explanation on that. The most similar problem is not necessarily recommended to a student. On a high level, if a student performs well on problems, we assume he/she performs well on similar problems as well, so we recommend a dissimilar problem and vice versa. More specifically, we project the performance of students on problems they solved onto the problems that they have not solved. This way, we have an evaluation of the performance of students on unseen problems. A problem is recommended that is within the capacity of students close to their boundary to help them learn, and at the same time recommendation is done so that all the concepts necessary for students are practiced by them.
> An evaluation on real students is presented in part 2 of the comment titled “Response to questions about Prob2Vec” on this page, and we observed that similar problems are more likely to be solved correctly at the same time or wrong at the same time.
> The math expressions are not ignored in our proposed Prob2Vec method. In the example given in the last paragraph on page 3 for example, math expressions are used to extract the concept n-choose-k. We both use math expressions and text to label problems with appropriate concepts.
>
> 2- Prob2Vec only uses expert knowledge for rule-based concept extractor, but does not use selected informative words. The effort put for rule-based concept extractor is negligible compared to effort needed for annotation of all problems with their corresponding concepts. We both annotated all problems manually and used rule-based concept extractor for annotation. In the former method, we observed 100% accuracy in the similarity detection test and observed 96.88% accuracy in the latter method. However, the rule-based concept extractor needs much less manual effort than manual problem annotation and is capable to provide us with relatively high level of accuracy we need in our application. Note that our method is scalable as long as problems are in the same domain as the rule-based concept extractor is automated for a single domain, but for the case that problems span many different domains, it is the natural complexity of the data set that requires a more sophisticated rule-based concept extractor. Furthermore, in most realistic cases for education purposes, problems span a single domain not multiple ones.
>
> We also like to grab your attention to the negative pre-training method proposed for training on imbalanced data sets. You may want to refer to part 2 of comment titled “Response to Question on Negative Pre-Training” and part 1 of our response to reviewer2.

---

### Official Review · AnonReviewer2 · 2018-11-01
**Proposes a method for mathematical problem embedding but the contribution is not strong**

**Rating:** 5
**Confidence:** 3

**Review:**

This paper proposes a method for mathematical problem embedding, which firstly decomposes problems into concepts by an abstraction step and then trains a skip-gram model to learn concept embedding. A problem can be represented as the average concept (corresponding to those in the problem) embeddings. To handle the imbalanced dataset, a negative pre-training method is proposed to decrease false and false positives. Experimental results show that the proposed method works much better than baselines in similar problem detection, on an undergraduate probability data set.
Strong points:
(1)	The idea of decomposing problems into concepts is interesting and also makes sense.
(2)	The training method for imbalanced datasets is impressive.
Concerns or suggestions:
1.	The main idea of using contents to represent a problem is quite simple and straightforward. The contribution of this paper seems more on the training method for imbalanced data sets. But there are no comparisons between the proposed training method and previous related works. Actually, imbalance data sets are common in machine learning problems and there are many related works. The comparisons are also absent in experiments.
2.	The experimental data set is too small, with only 635 problems. It is difficult to judge the performance of the proposed model based on so small data set.
3.	The proposed method, which decomposes a problem into multiple concepts, looks general for many problem settings. For example, representing a movie or news article by tags or topics. In this way, the proposed method can be tested in a broader domain and on larger datasets.
4.	For the final purpose, comparing problem similarity, I am wondering what the result will be if we train a supervised model based problem-problem similarity labels?

---

> ### Author Response · Authors · 2018-11-07
> **Response to Reviewer2**
>
>
> 1- The idea of using concepts to represent a problem is simple, but using it along with neural network based embedding gives us the opportunity to gain concept continuity as discussed on the last paragraph on page 7 and table 2, which is an active field of research in education.
>
> The focus of this work is on problem embedding and its application in a recommendation system that uses problem embedding to project students’ performance for the problems they solved onto the problems that they have not solved yet. Using the evaluation on unseen problems, a problem is recommended that is within the capacity of students close to their boundary to help them learn, and at the same time we cover all the concepts necessary for them to learn. In the meanwhile, we got the interesting idea of negative pre-training on training with imbalanced training data and tested our hypothesis and included in the paper. Due to space limit, we did not include the literature review and comparison of other methods in terms of memory use and training complexity, but you can find them in the response of a previous comment below titled “Response to Question on Negative Pre-Training” on this page to see the comparison. We can include the literature review for training on imbalanced data sets as well as comparison of other methods with negative pre-training in terms of memory use and training complexity in the final version. In summary, a) oversampling extremely suffers from over-fitting, b) SMOTE method that generates synthetic data sample is not feasible in word space, so the generated synthetic data (that are mathematical problems) are not of use for our training purpose, c) borderline-SMOTE both suffers from the same issue as SMOTE and its high complexity for finding the pairwise distance between all data samples, which is a burden in high dimensional data, and d) hybrid methods need m >> 1 weak learners in contrast to negative pre-training that uses a single learner. Memory use and training time is an issue for hybrid method when the weak learners are deep neural networks with too many parameters. We are currently running a broader experiment for negative pre-training on other data sets to gain more insight on it, but for the purpose of the task proposed in this work, it outperforms one-shot learning, which cannot be said that is the state-of-the art, but is a common practice. There is no notion of state-of-the-art in training on imbalanced data sets since due to our best knowledge, there is no method that outperforms all the other ones, and the performance of different methods depends more on the nature of the data set.
>
> 2- The data set being small is the nature of the application since creating mathematical problems is a creative process, so it is hard to have a very big data set. The Prob2Vec method is performing well on this not relatively big data set, which is our goal, but if we have a bigger data set (as we have right now with more than 2400 problems), Prob2Vec may even have a better performance since with more data we can have a more precise concept and problem embedding.
>
> 3- Thanks for your suggestion.
>
> 4- It is difficult for humans to determine a similarity score consistent across a large enough training set, so it is not feasible to simply apply supervised methods to learn a similarity score for problems. Even if problem-problem similarity annotation is feasible, a lot of effort should go into the annotation, which is not scalable.

---

### Official Review · AnonReviewer3 · 2018-11-03
**small technical contribution**

**Rating:** 3
**Confidence:** 3

**Review:**

The paper proposed a hierarchical framework for problem embedding and intended to apply it to adaptive tutoring. The system first used a rule-based method to extract the concepts for problems and then learned the concept embeddings and used them for problem representation. In addition, the paper further proposed negative pre-training for training with imbalanced data sets to decrease false negatives and positives. The methods are compared with some other word-embedding based methods and showed 100% accuracy in a similarity detection test on a very small dataset.

In sum, the paper has a very good application but not good enough as a research paper. Some of the problems are listed as follows:
1.	Lack of technical novelty.  It seems to me just a combination of several mature techniques. I do not see much insight into the problem. For example, if the rule-based concept extractor can already extract concepts very well, the “problem retrieval” should be solved by searching with the concepts as queries. Why should we use embedding to compare the similarity? Also, the title of the paper is about problem retrieval but the experiments are about similarity comparison, there seems a gap.
2.	Data size is too small, and the baselines are not state-of-the-art. There are some unsupervised sentence embedding methods other than the word-embedding based models.
Some clarity issues. For example, Page 6. “is pre-trained on a pure set of negative samples”— what is the objective function? How to train on only negative samples?

---

> ### Author Response · Authors · 2018-11-07
> **Response to Reviewer3**
>
>
> 1- There are two reasons that concept and problem embedding are performed in this work. Considering concept continuity is an important matter in education. Having concept embedding, concept continuity can be reached as is discussed in the last paragraph on page 7 and some other examples are given in table 2. By just having the most sophisticated concept extractor, the concept continuity cannot be retrieved. Furthermore, problem embedding is used by the recommender system to project the performance of students on the problems they solved onto other problems that they have not solved. This way, we have an idea of what problems should be recommended to them and which problems should not by having an evaluation of their ability to solve unseen problems and recommend problems in the boundary of their capacity, not way beyond, and to recommend problems in a way that covers all concepts necessary for students to learn. We have observed interesting patterns, e.g. similar problems are more likely to be solved correctly at the same time or wrong at the same time. Note that by just having the concepts of problems that are not in numerical form, performance projection may not be feasible and there is a need for using other methods like embedding.
>
> 2- The data size being small is just the nature of the application. Creating new problems is a creative process and is not easy, given that with the insight we have on the application, the data size seems to suffice. Furthermore, since Prob2Vec is performing well for not a relatively big data set, it would definitely do well for big data sets since the more data we have, the more precise the concept and problem embedding are. The easy-tough-to-beat method proposed by Arora et al. is the state of the art in unsupervised sentence embedding that we compared our algorithm with. Please let us know if we missed anything.
>
> Pre-training is a common practice in transfer learning (one-shot learning). The objective function does not differ from the objective function used for post training. Training on only negative samples with lower training epochs than the training epochs in post training just adjusts the weights of the neural network to a better starting point. If the training epochs in pre-training is relatively smaller than the training epochs in post training, due to curse of dimensionality, the warm start for post training results in better performance for NN classifier. To make it more clear what it means to train the neural network on a pure set of negative data samples, think about batch training. It's not likely, but possible, that a batch only has negative or positive samples. In the pre-training phase of our method, we intentionally used a pure set of negative samples (with fewer training epochs) to have a warm start for post training. As table 3 shows, our proposed method outperforms one-shot learning. Please look at part 1 of our response to reviewer2 and part 2 of comment titled "Response to Question on Negative Pre-Training" below.

---

### Public Comment · ~Mohammadamir_Kavousi1 · 2018-10-07
**Negative Pre-Training Details**

I have two questions on your proposed negative pre-training algorithm as follows:

1- Do you use the same number of training epochs for the first and second phases of negative pre-training? If yes, why, if no, what's the intuition behind it?

2- I know it's not applicable to compare the performance of your negative pre-training method with all other existing methods for classification with having imbalanced training data sets, and there is not such a notion of state-of-the-art algorithm for such methods, and probably the most prominent one is down sampling to avoid training complexity and over-fitting, but do you have any comparison of training complexity in terms of memory use and rough training time of negative pre-training and other algorithms for classification with having imbalanced training data sets?

---

> ### Author Response · Authors · 2018-10-10
> **Response to Question on Negative Pre-Training**
>
> Thanks for your comment. Here are the responses to your two questions:
>
> 1- The ratio of the number of training epochs in the first and second phases of the negative pre-training method is a hyper-parameter of this method. In our simulations, the number of training epochs in the first phase is half of those in the second phase. Note that if the number of training epochs in the first phase goes to zero, negative pre-training would become a pure down sampling. On the other hand, if the number of training epochs in the first phase is much larger than those in the second phase, the neural network cannot learn the structure of data in the second phase. Hence, we believe the ratio should not be large, but relatively small.
>
> 2- As you mentioned, it is not feasible to rank methods for classification over unbalanced data sets, but their complexity in memory use and training time can definitely be discussed. Based on our extensive literature review on classification on unbalanced data sets, we found the following methods that are compared in complexity with negative pre-training below:
>
> a) Under/Over sampling: under sampling (down sampling) has its own benefits of very low complexity and high speed, but as we see in our paper, the cost is low performance. Over sampling usually suffers from over-fitting specially when the imbalance in data set is high. In our case, if we want to use over sampling, we need to at least replicate each positive data sample for 50 times which is prone to extreme suffer from over-fitting. Negative pre-training obviously has more training time than under sampling (but gives better performance), but it needs about half memory and training time compared to over sampling (in case that over sampling is done to completely balance the training data set).
>
> b) SMOTE: this method generates synthetic data in order to bring balance for negative and positive data samples. For extreme imbalance in training data, this method can be prone to over-fitting as well. Regarding memory usage and training time, negative pre-training needs about half of those compared to SMOTE (in case that SMOTE is used to completely balance the training data set).
>
> c) Borderline-SMOTE: this method is in nature similar to SMOTE, but adds synthetic data in the border of the negative and positive sample. The method that is used to find the data samples in the border has high complexity, where the pairwise distance between the positive samples and all other samples should be measure (which can be hard for high dimensional data). Hence, although this method outperforms SMOTE, it needs strictly two times more memory and training time compared to negative pre-training, but for data with high dimension, it can be much worse or even impossible to find the pairwise distances between all data samples.
>
> d) Hybrid method: in this method, different weak learners are trained over the unbalanced training data set, then Adaboost method is used to combine the weak learners into a weighted sum that represents the boosted classifier. The comparison of hybrid method with negative pre-training in terms of memory usage and training time depends on how many weak learners we want to have and train. For m weak learners, we need to train m distinct neural networks, while we only have a single neural network in negative pre-training. Hence, this method is more complex than our proposed method and needs to store the weights of m neural networks that can be infeasible for deep networks (it is usually the case that m >> 1).

---

### Public Comment · (anonymous) · 2018-10-22
**General questions about Prob2Vec**

My background on natural language processing suggests that you could’ve also annotated similarity among a set of training problems and trained a supervised machine learning model to predict the similarity of the unseen problems in the test set. Do you have any ideas if this can result in a better or comparable performance to Prob2Vec in your similarity detection test?

Have you surveyed the performance of your proposed recommendation system based on Prob2Vec and fluency projection on problems based on their similarity scores to see how it works besides having good performance on the similarity detection test?

---

> ### Author Response · Authors · 2018-10-24
> **Response to questions about Prob2Vec**
>
> 1-	We believe that it is easier to keep consistency in concept labeling than similarity annotation for a set of training problems. Furthermore, concept labeling in Prob2Vec is automated by a rule-based concept extractor, where the rules for concept extraction are relatively easy to find for experts. However, similarity annotation requires much more expert effort to prepare a relatively large training data set. In general, it is difficult to determine a similarity score consistent across a large enough training set, so it is not feasible to simply apply supervised methods to learn a similarity score for problems.
>
> 2-	We divided the probability course into 26 modules, where each module is on a specific topic. About 300 students who practiced on our platform were asked about the performance of the recommendation system after they practiced for each module (some students practiced a module for more than once at their own will). Hence, we got around 7000 feedback, where about 76% of them had positive responses on the performance of the recommender system. Furthermore, we observed that similar problems are likely to be done correctly at the same time or wrong at the same time by students.

---

### Meta-Review · Area_Chair1 · 2018-12-17
**Lack of technical novelty**

**Confidence:** 5
**Recommendation:** Reject

**Metareview:**

I tend to agree with reviewers. This is a bit more of an applied type of work and does not lead to new insights in learning representations.
Lack of technical novelty
Dataset too small